# Research Progress in Preparation and Purification of Rare Earth Metals

**Hang Liu [1], Yao Zhang [2], Yikun Luan [1,\*], Huimin Yu [3] and Dianzhong Li [1]**

[1]  Institute of Metal Research, Chinese Academy of Sciences, Shenyang 110016, China; liuhang@imr.ac.cn (H.L.); dzli@imr.ac.cn (D.L.)
[2]  Baotou Rare Earth R&D Center, Chinese Academy of Sciences, Baotou 014000, China; zhangyao@btzkxt.cn
[3]  School of Metallurgy, Northeastern University, Shenyang 110819, China; hmyu@ire.ac.cn
[\*]  Correspondence: ykluan@imr.ac.cn

**Abstract:** The purity of rare earth metals is one of the most important factors to research and develop high technique materials. However, high purity rare earth metals are not easily achieved. This review summarizes the preparation and purification methods of rare earth metals. First, the preparation principle and process of molten salt electrolysis and metal thermal reduction are introduced. The main sources of metallic impurities and interstitial impurities in rare earth metals as well as the action mechanism of reducing the concentration of different impurities are analyzed and summarized. Then, the purification principle and process of vacuum distillation, arc melting, zone melting, and solid state electromigration are also discussed. Furthermore, the removal effect and function rule of metallic impurities and interstitial impurities in rare earth metals are outlined. Finally, the crucial issues in the development of high purity rare earth metals are put forward, and the development direction of high purity rare earth metals in future are pointed out on this basis.

**Keywords:** rare earth metals; purity; impurity concentration; preparation; purification

## 1. Introduction

Rare earth element (named as RE) is the general name of 17 special elements, containing lanthanide element, scandium, and yttrium. They have been widely applied in functional materials, steel, and nonferrous metals because of their special optical, electrical, and magnetic properties. In particular, RE functional materials and structure materials play an important role in the development of aerospace industry, military equipment [1], domestic appliances [2], new energy saving, and environmental technologies [3,4]. For example, neodymium and samarium are the main components of NdFeB permanent magnets and SmCo permanent magnets respectively. Furthermore, magnetic energy product of RE permanent magnet materials is much higher than that of ferrite and Al-Ni-Co permanent magnet materials [5,6]. Therefore, RE permanent magnet materials have been proverbially used in air conditioning, sound box, and permanent magnet motor. Terbium and dysprosium are the main ingredients of magnetostrictive materials, and the magnetostrictive coefficient is superior than that of Fe-Ni-Co alloy. In addition, RE magnetostrictive materials have been widely used in high-power emitting sonar, sensor, and communications. RE hydrogen storage alloy shows the advantages of high electric capacity, good stability, high hydrogen absorption efficiency, and no pollution, which makes it to be widely used in the fields of battery, brake, and refrigeration [7]. The addition of rare earth to steel can modify inclusions, refine grain, and strengthen microalloying, which can significantly improve the fatigue performance of bearing steel [8]. The purity of rare earth metals is the key factor affecting the performance of functional materials and structural material. For example, high oxygen concentration of rare earth metals could weaken the intrinsic coercivity of RE permanent magnet

materials. Low purity rare earth metals may cause nozzle clogging and unstable performance in continuous casting process of RE steel. Specially, magnetostrictive materials and sputtering target materials require the purity of rare earth metals to be higher than 99.99% [9,10]. The purity of rare earth metals should exceed 99.95% in permanent magnetic materials [11]. In recent years, the preparation and purification of high-purity rare earth metals has attracted more attention from governments and experts. The relevant departments of the United States, Japan, and other countries have even listed RE products as the key strategic elements for the development of military technology and high technique industries. With the progress of science and technology, high purity rare earth metals play a critical role in national economic construction and daily life [12,13].

## 2. Progress in Preparation of Rare Earth Metals

With the wide applications of rare earth metals, the increasing demand for high-purity rare earth metals has stimulated the rapid development of preparation technology. At present, molten salt electrolysis and metal thermal reduction are the common methods for preparing rare earth metals. In principle, these two methods can extract all kinds of rare earth elements. Light rare earth metals such as La, Ce, Pr, and Nd are produced by molten salt electrolysis for considering economic cost factors such as fixed asset investment, raw materials, and energy consumption [14]. Metal thermal reduction method is more suitable for preparing heavy rare earth metals such as Gd, Tb, and Y with high melting and boiling points.

### 2.1. Molten Salt Electrolysis

Several kinds of metals such as Cu, Al, and Mg are prepared by molten salt electrolysis [15–17]. The preparation of rare earth metals by molten salt electrolysis started from the molten salt system of chloride. Because of the problems of chloride electrolysis preparation such as easy moisture absorption of raw materials, low yield and serious exhaust pollution, the preparation of rare earth metals by rare earth chloride was clearly banned by the state [18]. Nowadays, rare earth metals are mainly prepared by fluoride oxide electrolysis in industry [19–21]. The schematic diagram of molten salt electrolysis is shown in Figure 1. $RE_2O_3$ dissolved in fluoride molten salt and dissociated into rare earth cations and oxygen anions. Under the action of direct current field, rare earth cations move toward cathode and can be reduced to metals by obtaining electrons [22]. Oxygen anions move to the anode and lose electrons, forming oxygen or interacting with graphite to form $CO_2$ and CO, the reactions that occur in this system are as follows [23]:

$$\text{-Cathode(reduction) reaction: } RE^{3+} + 3e^- = RE \tag{1}$$

$$\text{-Anode(oxidation) reaction: } 2O_2 - 4e^- = O_2(g) \tag{2}$$

$$\text{-Reduction-oxidation (Redox) reaction: } RE_2O_3 + C = 2RE + CO_2 + CO \tag{3}$$

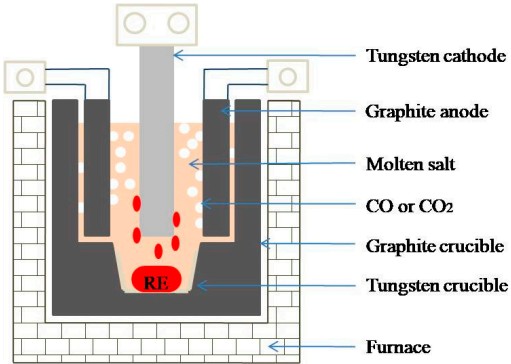

**Figure 1.** Schematic diagram of molten salt electrolysis.

Metallic impurities such as Al, Si, Ti, W, Mo, Fe and interstitial impurities C, O, and N might exist in rare earth metals prepared by the above method [24,25].

Al and Si impurities mainly are derived from furnace materials [26] and raw materials [27]. The concentration of Al and Si in rare earth metals were reduced from 0.03% to below 0.01% by altering the variety of furnace materials and the purity of raw material [26]. Carbon impurity is mainly related to graphite anode or graphite crucible, and iron impurity mainly came from anode clamp and auxiliary tools [28]. By adjusting electrolytic bath structure, electrolytic temperature and other technical parameters can control the carbon concentration in rare earth metals below 0.03% and iron concentration below 0.3% effectively [28,29]. Moreover, W, Ti, and Mo impurities are mainly derived from metal crucible and stirring device. The total concentration of W, Ti, and Mo could be reduced to below 0.05% by adjusting the crucible and stirring device materials [30]. Impurities of O and N are mainly derived from rare earth oxide and the oxygen in air. The oxygen concentration of commercial electrolysis rare earth metal was even as high as 0.1% [31]. In recent years, researchers of IMR have explored the oxygen reduction methods in molten salt electrolysis process, and high purity rare earth metals were successfully prepared by regulating technical parameters, as shown on in Figure 2. At the same time, oxygen concentration could be reduced to less than 0.008% after using suction casting and airless shot blasting, as shown in Figure 3 [32].

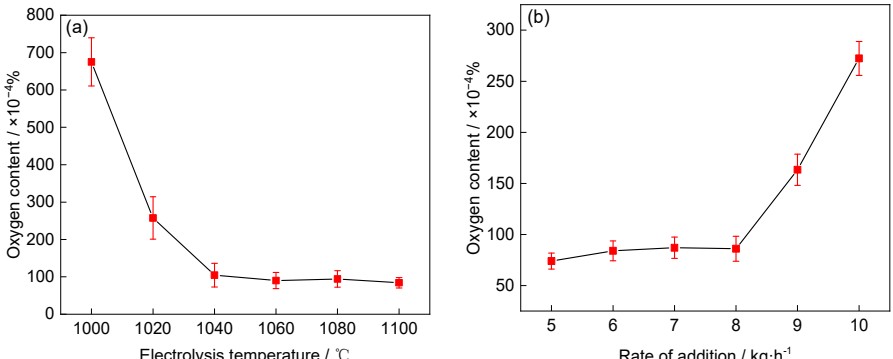

**Figure 2.** Effect of electrolysis temperature (**a**) and rate of addition (**b**) on oxygen concentration.

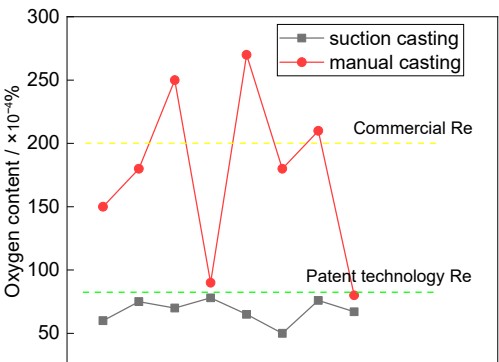

**Figure 3.** Effect of casting method on oxygen concentration.

Baotou Research Institute of Rare Earths successfully developed a 3 kA fluorosalt system electrolytic bath in 1984. After more than 30 years of development, the scale of electrolytic bath has gradually expanded from 3 to 10 kA [33], and the maximum electrolytic bath could reach 60 kA [34,35]. So far, large-scale rare earth enterprises such as Ruixin Group and Qiandong Group have already used 10 kA-level electrolytic bath [36]. However, because of the low level of automation, most of the rare earth enterprises are still using 6~10 kA electrolytic bath [37].

### 2.2. Metal Thermal Reduction

Metal thermal reduction refers to the reduction of rare earth compounds into metals at high temperatures by active metal reducing agents, such as Ca, La, and Ce [38]. Sm, Eu, Yb, and Tm have moderate melting point and relatively high vapor pressure, so La or Ce with relatively low vapor pressure are used as reductants, the reduction equation is as follows:

$$2La + RE_2O_3 = 2RE\uparrow + La_2O_3 \tag{4}$$

Y, Gd, Tb, Dy, Ho, Er, Lu are prepared by calcium thermal reduction, the reduction equation is as follows:

$$3Ca + 2ReF_3 = 3CaF_2 + 2RE \tag{5}$$

Metallic impurities such as Ca, Ni, Mo, Ti, W and interstitial impurities C, O, and N might exist in rare earth metals prepared by the above method [39]. The schematic diagram of metal thermal reduction method is shown in Figure 4.

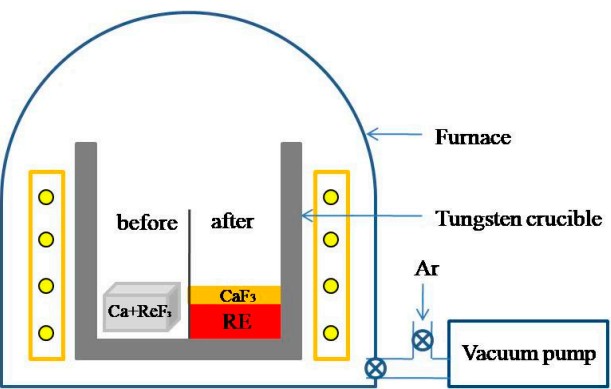

**Figure 4.** Schematic diagram of metal thermal reduction.

Impurities of rare earth metals are mainly derived from raw materials. Ni impurity mainly came from the nickel boat. Ni concentration could be reduced from 0.01% to 0.003% by reducing the temperature of the fluorination process or applying a protective layer on the nickel boat [40]. Impurities such as Si, Fe, Al, Mo, Ti were mostly from rare earth oxide and reagent materials. Total concentration of impurities in rare earth metals could be reduced from 0.12% to less than 0.01% by improving the purity of raw materials as well as using analytical reagent [41]. Oxygen impurity was mainly derived from raw materials, reducing agents, and oxidation of rare earth metals [42,43]. Oxygen concentration could be reduced from 0.1% to less than 0.07% by improving the purity of raw materials [40]. Impurities of W and Nb were mainly from metal crucible material. Tungsten concentration of rare earth metals could be reduced to less than 0.008% by improving vacuum remelting or vacuum distillation process conditions [39].

Up to now, single furnace output of metal thermal reduction has gradually developed from 100 g to 100 kg [44]. Its advantages are low investment and compact process, but there are also some issues such as harsh requirements on equipment, discontinuous production process, and generally higher impurity concentration than electrolytic rare earth metals.

In conclusion, the researches mainly focused on two aspects: purity and yield. Purity is mainly concerned with reducing metallic impurities such as Fe, Mg, and Si. The concentration of impurities can be reduced to a certain extent by adjusting the material and technical parameters of the equipment. However, there are fewer studies on effectively reducing the concentration of interstitial impurities such as O and N, and mechanism of action remain to be further studied. Yield is mainly concerned with low energy consumption and high recovery rate. The gas tightness of equipment and automation degree in preparation process should be more concerned in future.

### 3. Progress in Purification of Rare Earth Metals

Whether molten salt electrolysis or metal thermal reduction, the purity of rare earth metals is in a range of 95.5~99.5%, which cannot meet the requirements of high performance materials [45]. At present, the purification methods mainly include vacuum distillation, arc melting, zone melting, and solid state electromigration [46,47].

#### 3.1. Vacuum Distillation

In the process of metal thermal reduction, excessive reducing agent is usually added to confirm complete reaction. Therefore, rare earth metals prepared by this method have high impurity concentration and should be purified before application. Vacuum distillation was widely used in industry to remove reductant metals in order to obtain high purity rare earth metals [48–51]. Vacuum distillation means that rare earth metals are separated from impurities under high temperature and high vacuum. Figure 5 shows schematic diagram of vacuum distillation. Vacuum distillation requires that matrix metal should have high saturated vapor pressure and vapor pressure difference from the impurity element. Figure 6 shows the vapor pressures of rare earth elements and impurities at 1600 °C [52].

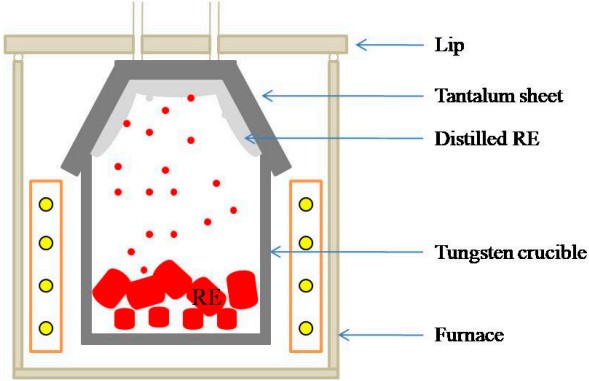

**Figure 5.** Schematic diagram of vacuum distillation.

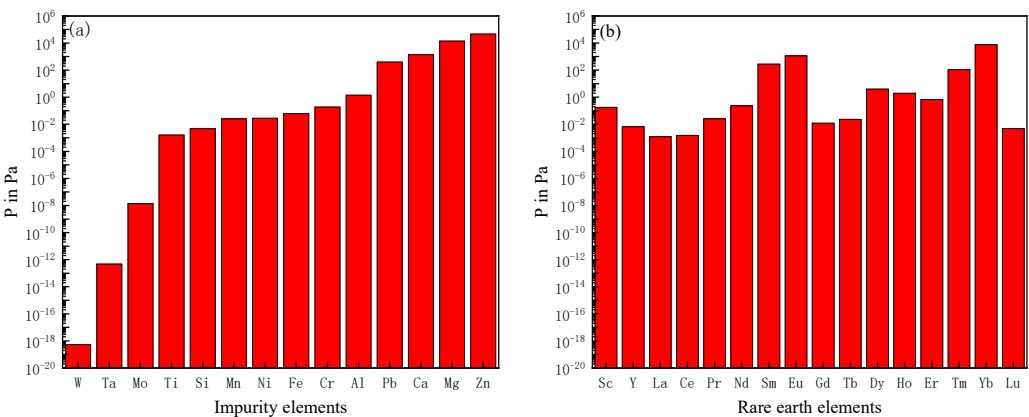

**Figure 6.** Vapor pressures of impurities (**a**) and rare earth elements (**b**) at 1600 °C.

Vacuum distillation can remove most metallic impurities and interstitial impurities. In the process of distillation, metallic impurities with high vapor pressure such as Ca, Mg, and Al have obvious removal effect. For example, total concentration of metallic impurities in Tb such as Ca, Mg, Al, Si, Zn, Cr, and Mn could be reduced from 0.065% to less than 0.001% [53]. Metallic impurities such as Ta, W, Mo, Nb, and Ti and interstitial impurities such as C, O, and N that exist in the form of compounds remained at the bottom of tungsten crucible [54–56]. This was mainly because the vapor pressure

of the impurities is much lower than that of the rare earth metals. W concentration of Gd could be reduced from 0.018% to 0.005% after vacuum distillation [57]. Interstitial impurity concentration of Sm was reduced to 0.0045%, and that of Yb reduced to 0.0051% [58]. The removal effect of metallic impurities with small vapor pressure difference from matrix metal was not obvious, such as Fe and Cu.

Vacuum distillation has the advantages of relatively high production efficiency and yield, but impurities segregation is prevalent in prepared rare earth metals [59]. Rare earth metal purification by vacuum distillation is still in the stage of small batch trial production, and the single furnace output is only 1~5 kg [60].

### 3.2. Arc Melting

Arc melting technology is a common method to purify metals and alloys. It uses electric energy to generate electric arc between electrode and molten material to melt the metal. Metallic impurities with high saturated vapor pressure and interstitial impurities can be effectively removed by using plasma arc as heat source to melt metal materials. At present, the working gas used in this method is mainly Ar, $H_2$, $N_2$, He and the mixture of above gases. In recent years, researchers have found that using the mixture of Ar and $H_2$ as the working gas could significantly improve the purification effect of rare earth metals. This melting method is called hydrogen plasma arc melting. In the present work, it can be proved that when the temperature reaches 5000 K, the dissociation of $H_2$ is up to above 95% [61,62]. The activated hydrogen atoms could combine with oxygen and nitrogen. The reactions that occur in this system are as follows:

$$H_2 = H + H \tag{6}$$

$$O \text{ (in metal)} + 2H = H_2O \tag{7}$$

$$N \text{ (in metal)} + 3H = NH_3 \tag{8}$$

This method has been successfully applied to the purification of rare earth metals such as La, Ce, Tb, and Gd [63,64]. Figure 7 is the schematic diagram of hydrogen plasma arc melting.

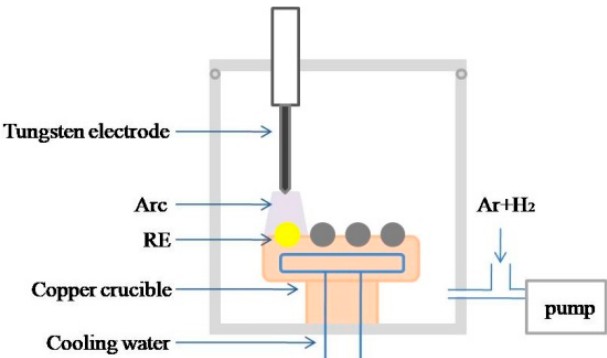

**Figure 7.** Schematic diagram of hydrogen plasma arc melting.

Metallic impurities with high saturated vapor pressure such as Ca, Mg, and Al could be effectively removed by arc melting. For example, total concentration of metallic impurities of Tb decreased from 0.438% to 0.0580% after arc melting [63]. Hydrogen plasma has high chemical reactivity. It can react with carbon, oxygen, nitrogen, and other impurities to form gaseous compounds, thus promoting impurities clearance [65]. When Tb and $LaNi_5$ alloy were melted in hydrogen plasma, carbon concentration of Tb decreased from 0.03% to 0.0017% [63], and carbon concentration of $LaNi_5$ alloy decreased from 0.018% to 0.0014% [64]. The concentration of O and N of Tb decreased from 0.05% to 0.0015% [66].

Hydrogen plasma arc melting have a good removal effect on metallic impurities and interstitial impurities. The removal degree of interstitial impurities is obviously better than other methods. However, because of high temperature and high vacuum required by the process, hydrogen plasma arc

melting technology for rare earth metals is still in the laboratory research stage, and its single furnace output is only 10~200 g [67,68].

### 3.3. Zone Melting

Zone melting is a process of redistributing impurities by making use of their different solubility in solid and liquid phases. In this operation, the researchers slowly melt the metal bar by moving the induction coil from one end of the metal bar to the other. This process allows multiple directional movements to accumulate impurities at both ends of the metal bar. Equilibrium distribution coefficient refers to the ratio of solute concentration in the solid and liquid phases when the solid and liquid systems reach equilibrium [69], and defined as:

$$K = C_S/C_L \tag{9}$$

where $C_S$ and $C_L$ are denoted as impurity concentration in solid phase and impurity concentration in the liquid phase respectively. Impurities whose $K < 1$ move toward the end of the metal bar, and impurities whose $K > 1$ move toward the beginning end of the metal bar. Figure 8 is the schematic diagram of plasma arc zone melting.

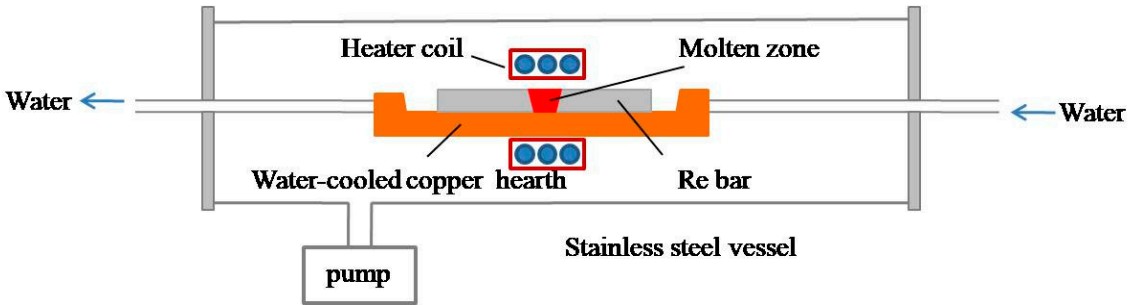

**Figure 8.** Schematic diagram of plasma arc zone melting.

Zone melting has been widely used to purify semiconductor materials such as Si, Sb, Te, and metal materials such as Zn and Cu [70]. Researchers have applied zone melting to purify rare earth metals, and the results showed that there were obvious segregation in the rare earth metals [71]. Metallic impurities with high saturated vapor pressure such as Ca, Mg, Zn, Mn could be reduced to less than 0.0001%. Fe, Cu, Al, Si could be enriched at the end as the melted zone moves [72]. Interstitial impurities such as O and N were enriched at the beginning [73]. Impurities such as C and S showed different migration patterns in different rare earth metals. After zone melting, impurities such as C and S of La were enriched at the beginning, while C and S of Ce were enriched at the end [73].

Zone melting had obvious migration effect on some metallic impurities and interstitial impurities. However, different migration rules of impurities usually appeared in different rare earth metals, and the purification mechanism needs to be further studied. The principle of zone melting is the redistribution of impurities rather than actual removal of impurities. In order to achieve excellent purification effect, nearly 20 times of repeated melting of rare earth metals are needed. Moreover, the purification efficiency of zone melting is low, and the single furnace output of rare earth metals is only about 30 g. Therefore, zone melting is only used as the final method to purify small batches of high purity rare earth metals [73,74].

### 3.4. Solid State Electromigration

Solid state electromigration means that under high vacuum conditions, the metal bar is placed between two electrodes and applied with direct current for a long time. The joule heat make the temperature of the test bar reach 0.8~0.9 times of the metal melting point and keep the test bar from

melting. Under the action of direct current, atoms are not neutral. In general, one atom can acquire more or less positive charge than the other, resulting in different ionization behaviors. Moreover, atoms can be affected by electrostatic force, electron collision friction and cavity friction [75], and defined as:

$$F_S = eEZ^0 \tag{10}$$

$$F_e = -eEn_e l_e \sigma_e \tag{11}$$

$$F_h = eEn_h l_h \sigma_h \tag{12}$$

$$F = F_S + F_e + F_h = eEZ^0 - eEn_e l_e \sigma_e + eEn_h l_h \sigma_h = eEZ^* \tag{13}$$

where e is elementary charge in units of C; E is electric field intensity in units of V·m$^{-1}$; $Z^0$ is the ion valence; n is unit volume electron; l is the electron mean free path; σ is the electron scattering cross section; Z* is effective valence.

Atoms with more positive charge migrate to the cathode extreme, and atoms with more negative charge migrate to the anode extreme. In the purification process, the migration rate of impurity atoms is higher than that of matrix atoms, so the impurities can be redistributed in matrix metal. Metals exhibit different crystal structures at different temperatures. Interstitial impurities show different migration patterns in different crystal structures. For example, when Pr is in double hexagonal closest packed structure (named as DHCP), H, N, and O atoms can enter the intercellular space except C. However, when Pr is transformed into BCC structure, all atoms cannot enter the intercellular space except H [76]. Therefore, researchers attempt to remove interstitial impurities by solid state electromigration at crystal temperature with a high migration rate of interstitial impurities [77,78]. Figure 9 is the schematic diagram of rare earth prepared by solid state electromigration.

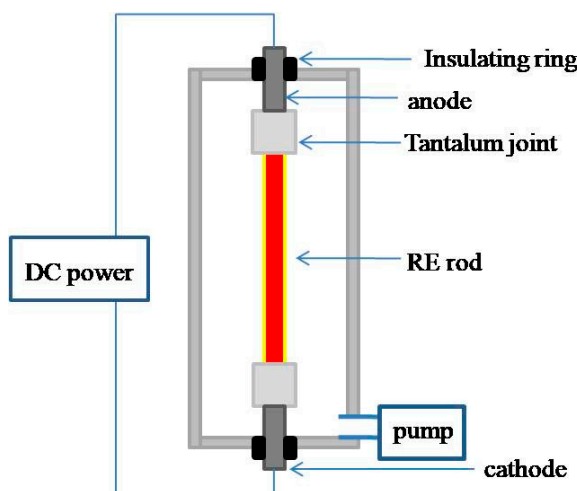

**Figure 9.** Schematic diagram of solid state electromigration.

For example, after 216 h of high vacuum and high temperature electromigration, the concentration of impurities such as Al and Si of Pr decreased from 0.04% to 0.02%, and total concentration of Fe and Ni decreased from 0.05% to 0.002% [79]. The concentration of interstitial impurities was reduced from 0.0239% to 0.0049%, among which the oxygen concentration was reduced to 0.0022% [80]. Solid state electromigration had significant removal effect on most metallic impurities and interstitial impurities [81], but not obvious removal effect on carbon and nitrogen impurities [82,83]. Solid state electromigration was essentially a process of impurities redistribution, and it did not really remove impurities, which will seriously affect the subsequent use of rare earth metals. Since the solid state electromigration process required high vacuum degree and energy consumption of equipment. It is

difficult to achieve large-scale industrial production, so this method is mainly used as the final purification method to produce small amount of ultra-high purity metals [84].

From the above, it is not possible to remove all impurities of rare earth metals simultaneously by any of the above methods. Therefore, after comprehensive consideration of the impurity types, purity requirements and removal effect of impurities, it should be considered that several purification methods should be combined to remove the impurities. For example, vacuum distillation combined with arc melting can improve the density and purity of rare earth metals. Ultralow oxygen concentration of rare earth metals can be prepared by the combination of arc melting and solid state electromigration.

## 4. Summary and Prospect

In the past 60 years of development, remarkable achievements have been achieved in scientific research and industrial application of rare earth elements, forming a relatively well-developed rare earth industrial system. But at the same time, it should also be noted that there are issues in many areas that need to be solved.

(1)  Molten salt electrolysis and metal thermal reduction possess high efficiency and yield, but low purity of rare earth metals and high energy consumption are generally non-negligible.
(2)  The preparation process of rare earth metals is mainly a manual operation with low automation degree and severe environmental pollution issue.
(3)  Deep purification ability of rare earth metals is insufficient, and the industrialization is limited to the use of vacuum distillation, which is difficult to meet the needs of high technology development.

Consequently, expensive price of high-purity rare earth metals appeared and limit their application. Therefore, the following directions are the focus of our attention in future:

(1)  The impurity concentration of rare earth metal should be further reduced so as to provide high purity starting material for the purification process.
(2)  A combination of various purification methods should be chosen to remove the impurities for considering the kinds and uses of rare earth metals.
(3)  Purification methods suitable for mass production such as vacuum levitation melting and electron beam melting should be developed and researched.

**Author Contributions:** Conceptualization, H.L. and Y.L.; methodology, H.L. and Y.L.; software, H.L.; validation, H.Y.; formal analysis, H.L.; investigation, H.L.; resources, D.L.; data curation, H.Y.; writing—original draft preparation, H.L.; writing—review and editing, H.Y.; visualization, Y.Z.; supervision, Y.L. and D.L.; project administration, Y.Z.; funding acquisition, Y.L. All authors have read and agreed to the published version of the manuscript.

**Funding:** This research was supported by Strategic Priority Research Program of the Chinese Academy of Sciences funded by Chinese Academy of Sciences (ACS) (XDC04010200). This work was also supported by Inner Mongolia Science and Technology Major Project funded by Inner Mongolia Science and Technology Department (2017B001).

**Conflicts of Interest:** The authors declare no conflict of interest.

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
