# Peer review of "Research Progress in Preparation and Purification of Rare Earth Metals"

_metals, doi:10.3390/met10101376_

Round 1

Reviewer 1 Report

I would like to submit some suggestion for consideration before publication.

  1. English needs to be improved. The manuscript is difficult to understand.
  2. There are a lot of typographical error in the text.
  3. The introduction is too short. A general overview of the topic should be provided. Furthermore, 6 references support one sentence (lines 29 and 30).
  4. Some details of the processes are missing. There are no reactions to clarify the processes.
  5. Figures 2 and 3 are not mentioned in the text. How were these figures obtained?
  6. The last section is too short and does not offer a prospect.
  7. Many references are in chinese and they are difficult to check.

Reviewer 2 Report

The removal effect and function rule of 19 metallic impurities and interstitial impurities in rare earth metals are outlined. Finally, the main 20 issues in the development of high purity rare earth metals are put forward, and the development 21 direction of high purity rare earth metals in future are pointed out on this basis.

The paper is very good. The paper can be published as presented.

Author Response

Dear reviewer:

Many thanks for your positive comments .It is my great honours receiving your recommendation.

Best regards!

Yours sincerely,

Reviewer 3 Report

This review article is interesting and well written. It clearly and succinctly presents the practical knowledge related to the separation and purification of rare earths. I would also like to highlight the efforts made by the authors to provide nice diagrams describing the different technologies presented. The only point which in my opinion deserves a little more detail concerns the efficiency of the processes presented for the separation of rare earths from one another. Indeed, the authors correctly discuss the separation of rare earths and other metals, however there are few details regarding the separation of different rare earths. 

"3.2 Arc smelting" replace by "3.2 Arc melting".
